# UltraVideo: High-Quality UHD Video Dataset with Comprehensive Captions

**Zhucun Xue**[1*]  **Jiangning Zhang**[1*†] **Teng Hu**[2] **Haoyang He**[1] **Yinan Chen**[1] **Yuxuan Cai**[3]
**Yabiao Wang**[1] **Chengjie Wang**[2] **Yong Liu**[1‡] **Xiangtai Li**[4] **Dacheng Tao**[4]

[1]Zhejiang University    [2]Shanghai Jiao Tong University
[3]Huazhong University of Science and Technology    [4]Nanyang Technological University
Project Page:  https://xzc-zju.github.io/projects/UltraVideo/

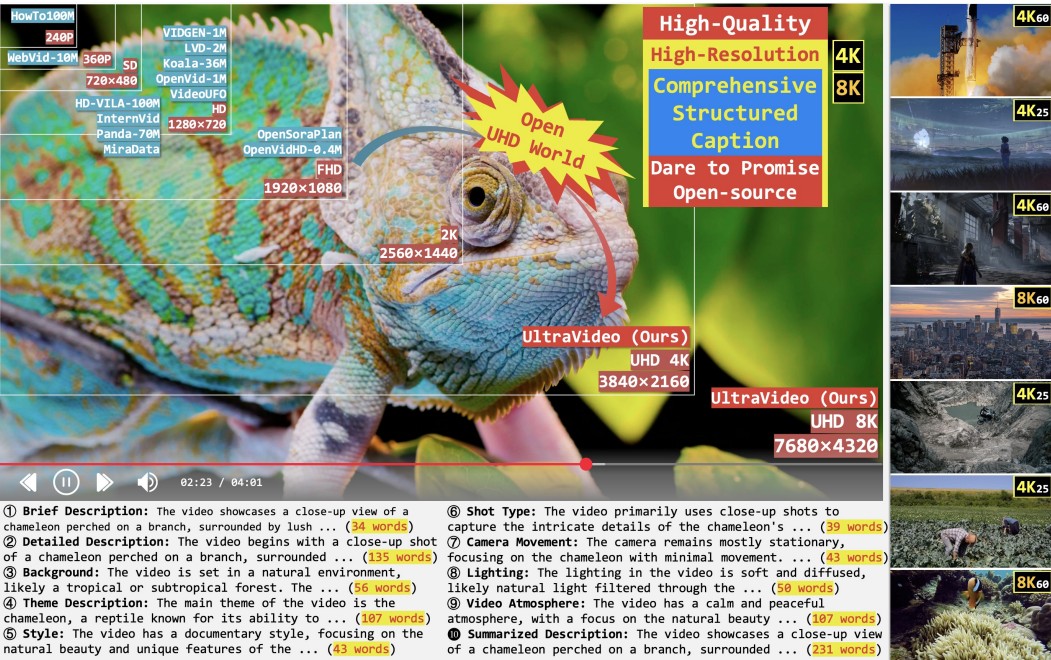

Figure 1: **UltraVideo** has *higher visual quality* and *ultra-high resolution* ($\geq$ 4K), along with *comprehensive structured captions* (bottom). Compared with current text-to-video (T2V) datasets, it can meet the growing demand for native high-resolution T2V applications. Benefiting from the carefully designed curation process, this dataset contains *diverse visually attractive scenes.* with the right side showing random samples with different resolutions and frame rates.

## Abstract

The quality of the video dataset (image quality, resolution, and fine-grained caption) greatly influences the performance of the video generation model. The growing demand for video applications sets higher requirements for high-quality video generation models. For example, the generation of movie-level Ultra-High Definition (UHD) videos and the creation of 4K short video content. However, the

---

*Equal contributions.

†Project lead.

‡Corresponding author.

39th Conference on Neural Information Processing Systems (NeurIPS 2025) Track on Datasets and Benchmarks.

existing public datasets cannot support related research and applications. In this paper, we first propose a high-quality open-sourced UHD-4K (22.4% of which are 8K) text-to-video dataset named UltraVideo, which contains a wide range of topics (more than 100 kinds), and each video has 9 structured captions with one summarized caption (average of 824 words). Specifically, we carefully design a highly automated curation process with four stages to obtain the final high-quality dataset: *i)* collection of diverse and high-quality video clips. *ii)* statistical data filtering. *iii)* model-based data purification. *iv)* generation of comprehensive, structured captions. In addition, we expand Wan to UltraWan-1K/-4K, which can natively generate high-quality 1K/4K videos with more consistent text controllability, demonstrating the effectiveness of our data curation. We believe that this work can make a significant contribution to future research on UHD video generation. UltraVideo dataset and UltraWan models are available at project page.

## 1 Introduction

The rapid development of video generation models has driven the continuous growth of the demand for high-fidelity and high-resolution content in fields such as film production, immersive media, and interactive entertainment [20]. However, the performance of text-to-video (T2V) models is severely limited by the quality of training data, especially regarding visual resolution, temporal consistency, and fine-grained semantic alignment with text descriptions. Although existing large-scale T2V datasets are abundant in quantity, they mainly focus on medium and low-resolution content (such as 720p) and simple captions, failing to meet the requirements for generating Ultra-High Definition (UHD) videos (4K/8K) with sharp details, rich textures, and precise semantic control [8, 36].

High-resolution video generation faces two core challenges. *Firstly*, resolution scalability: Models trained on low-resolution data generally struggle to generalize to UHD scenarios, and issues such as artifacts, blurriness, and inconsistent content are likely to occur when extrapolating to higher resolutions. As shown in Fig. 2, when the Wan-T2V model is directly applied to a 4K resolution without specialized training, the generation quality significantly deteriorates. *Secondly*, semantic granularity: Precise control over visual attributes (such as camera motion, lighting, style) requires structured captions that explicitly describe the scene semantics. However, most datasets lack comprehensive annotations that can guide the generation of such details.

To fill these gaps, we propose UltraVideo, a high-quality, open-source UHD-4K/8K T2V dataset designed to enhance the technical level of high-resolution video generation. This dataset contains 42K short videos (3~10 seconds) and 17K long videos (≥10 seconds). It is the first public dataset that gives priority to **native UHD resolution** and **structured captions**, which include 10 types of semantic tags (such as shot type, lighting, video atmosphere), with an average of 824 detailed words per video. The high quality of UltraVideo benefits from a four-stage data curation process: 1) Diverse clip collection: Screen videos with a resolution of ≥4K and a frame rate of up to 60FPS from YouTube, and exclude low-quality content through manual quality inspection (Sec. 2.1). 2) Statistical filtering: Remove videos with excessive text, black borders, abnormal exposure, or low saturation to ensure the purity of visual inputs (Sec. 2.2). 3) Model-based data purification: Utilize a large multimodal model (Qwen2.5-VL-72B [2]) to detect low-quality attributes (watermarks, captions) and quantify aesthetic and motion consistency to further refine the dataset. 4) Comprehensive Structured Caption: Use an open-source MLLM (Qwen2.5-VL-72B [2]) to automatically generate nine categories of detailed captions, supporting fine-grained semantic control during the training process (Sec. 2.4), and further use an LLM to generate detailed descriptions. To verify the effectiveness of UltraVideo, we extend the Wan-T2V model to UltraWan-1K/-4K, which is capable of natively generating high-quality 1K and 4K videos and improves text controllability. By optimizing the training strategy, it achieves advanced performance in UHD generation tasks, and still performs excellently even with a moderate dataset size (42K samples). In summary, our contributions are threefold:

*1)* To support the increasingly developing high-resolution video generation applications and bridge the gap between academic and large corporate data, we curate a high-quality UHD UltraVideo dataset, focusing on fine-tuning fundamental high-resolution video generation models with fine-grained structured captions.

*2)* With manually filtered video sources, we propose a sophisticated automated data processing pipeline, which includes high-quality data collection, filtering, and fine-grained structured captions.

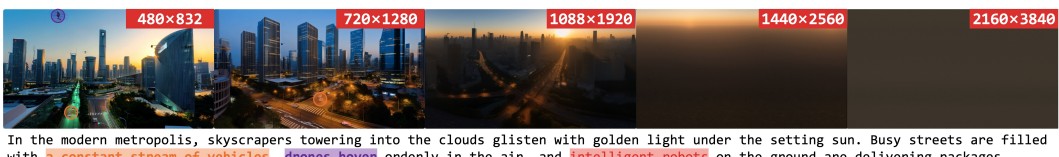

In the modern metropolis, skyscrapers towering into the clouds glisten with golden light under the setting sun. Busy streets are filled with a constant stream of vehicles, drones hover orderly in the air, and intelligent robots on the ground are delivering packages.

Figure 2: Wan-T2V-1.3B [34] shows a significant decline in visual quality and semantic consistency as the resolution increases, and it fails to generate high-resolution videos without.

*3)* Based on Wan-T2V-1.3B, we have improved the high-resolution generation architecture UltraWan and proposed a caption sampling strategy. Through fine-tuning with LoRA plugins, it can support the generation of videos with native UHD resolution. The results of evaluations by VBench and human assessments have demonstrated its superiority.

## 2 Curating UltraVideo Dataset

Recent T2V datasets emphasize the quantity of videos (million-level 720p videos) with detailed captions that can support the pre-training of video models. In contrast, we mainly focus on the quality of the UHD video dataset we construct for high-quality model fine-tuning, i.e., high-quality image quality, high-resolution frames, and comprehensive captions. Considering that mainstream video generation models only support video generation for a few seconds, for example, HunyuanVideo [15] supports a maximum of 129 frames and Wan [34] supports 81 frames. This paper mainly focuses on the construction and evaluation of short videos. Of course, we also open-source the affiliated long videos for the increasingly popular long video duration generation with the same processing flow. Fig. 3 intuitively outlines the specific data curation pipeline, which contains four steps: 1) Video Clips Collection (Sec. 2.1). 2) Statistical Data Filtering (Sec. 2.2). 3) Model-based data purification (Sec. 2.3). 4) Comprehensive Structured Caption (Sec. 2.4).

### 2.1 Video Clips Collection

**UHD-4K/8K video source.** Most of the recent popular text-to-video datasets are directly or indirectly sourced from the HD-VILA-100M dataset [8, 20, 31, 36], while MiraData [14] has collected 173K video clips from 156 selected high-quality YouTube channels. We believe that for a high-quality video dataset, strict control should be exercised at the source of collection, which can strictly limit the number of videos entering the filtering process. The benefits of this approach are obvious. It can reduce the computational power and storage pressure during the screening process. At the same time, it can reduce the proportion of low-quality data and improve the quality of the final dataset. To this end, we have decided to use the 4K/8K video pool on YouTube as the sole source. The selected videos consist of two parts: 1) First, from the filtered Koala-36M [36] dataset, a subset is obtained by screening based on resolution (greater than 4K), frame rate (higher than 25FPS), and duration (longer than 30 seconds), and videos that users are not interested in are screened out through meta user behavior information (views, likes, and comments). Furthermore, by calculating the similarity between the video titles and descriptions and the pre-classified video themes, the highest-quality videos of each category are uniformly sampled and duplicates are removed. 2) We use large language models (LLMs) to generate some relevant recommended search keywords according to 108 themes, and manually search for the latest 4K/8K videos related to these themes. Eventually, we obtain 5K original videos, with lengths ranging from 1 minute to 2 hours. And we conduct a secondary manual review of these videos to ensure as much as possible that there are no problems such as low quality, blurriness, watermarks, and jitter to obtain high-quality original videos.

**Video theme.** The theme diversity of videos is crucial for the training effect of video models. Therefore, we conducted a noun statistics on the captions of Koala-36M. The results were processed by a large language model (LLM), and finally, through manual post-modification and confirmation, we obtained seven major themes (108 topics), namely: i) video scene, ii) subject, iii) action, iv) time event, v) camera motion, vi) video genres, and vii) emotion. Fig. 4 has statistically analyzed the proportion of clips for different topics under each theme. It can be seen that our UltraVideo contains diverse themes.

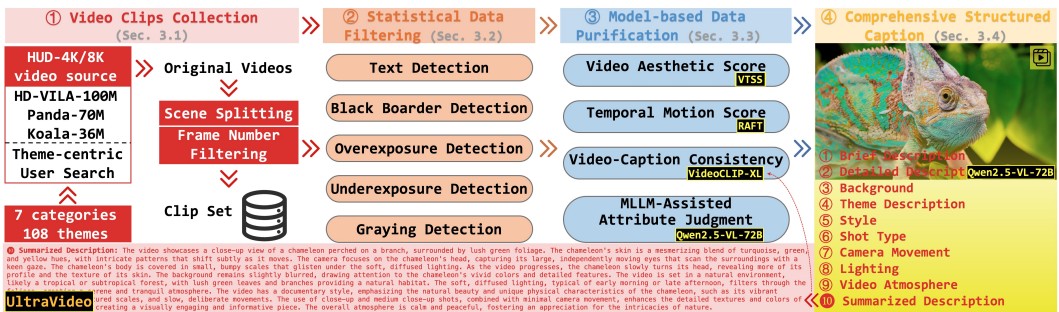

Figure 3: **Our video curation process that includes four data collection processes**: ① Video Clips Collection (Sec. 2.1), ② Statistical Data Filtering (Sec. 2.2), ③ Model-based data purification (Sec. 2.3), and ④ Comprehensive Structured Caption (Sec. 2.4). Ultimately, we obtain 42K high-quality UHD short clips with comprehensive descriptions.

**Scene splitting.** We use the popular PySceneDetect [5] to segment the original video into clips. Specifically, a two-pass AdaptiveDetector detector is employed, which applies a rolling average to help reduce false detections in scenarios like camera movement. In addition, we found that this detector might overlook videos with dissolve transitions. Therefore, we use DINOv2 to calculate the feature similarity for the first and last 5 frames of each video to further filter the videos.

**Frame number filtering.** Mainstream video generation models only support video generation for a few seconds. For example, HunyuanVideo [15] supports a maximum of 720×1280 resolution with 129 frames, while Wan [34] supports 720×1280 resolution with 81 frames, and the average video length of most video datasets is less than ten seconds. However, there has been a recent trend in long video generation research. For instance, MiraData [14] focuses on long duration video generation. Taking the above two points into account, we first filter videos with a time length between 3 seconds and 10 seconds as the short video set, and videos with a frame duration of more than 10 seconds are regarded as the long video set to support future research related to long videos (this setting will not be discussed in detail in this paper). To further expand the number of short videos, for long videos with a length of less than 60 seconds, we take the middle 10 seconds as short videos, and for videos longer than 60 seconds, we additionally take 10 seconds of video from both sides as short videos. Finally, we obtained 62K short videos with a duration of 3 seconds to 10 seconds and 25K long videos with a duration of 10 seconds or longer.

## 2.2 Statistical Data Filtering

At the statistical level, we conduct a secondary strict filtering of the videos by setting a mean threshold.

**Text detection.** Text inevitably appears in different time intervals of the original video. Large areas of text usually include subtitles, logos, and other markings. An excessively high proportion of such text can have a negative impact on model training. We use PaddleOCR [22] to detect text in each frame and calculate the proportion of the union area of the minimum bounding rectangles of all detected text within the frame to the total image area. If this proportion exceeds a strict threshold of 2%, the frame is considered problematic. Finally, we calculate the ratio of problematic frames to the total number of frames and rigorously exclude videos with a ratio higher than 5%.

**Black border detection.** Black borders often appear in movies and user-edited videos. We calculate the mean value of the rectangular area that extends from the four sides towards the middle by 3%. If the calculated value is lower than 3, the frame is regarded as an abnormal frame. Finally, we calculate the proportion of the number of problematic frames to the total number of frames, and if it is higher than 5%, the video will be excluded.

**Exposure detection.** Overexposure and underexposure greatly affect the video image quality. Taking 5 as the threshold, we calculate the proportion of pixels that are higher than 250 and lower than 5 for each frame. If the proportion is higher than 12%, the frame is considered to have a problem. We remove videos with more than 5% of bad frames.

**Graying detection.** Images that are grayish or have low saturation often give people an unpleasant visual experience. We calculate the variance of RGB values at each position and then take the average

value for the entire image. If this average value is lower than 1.2, the frame is considered to have a problem. Similarly, if the proportion of such frames in the whole video is higher than 5%, the video will be removed. At this stage, we obtained 46K short videos with a duration of 3 seconds to 10 seconds and 19K long videos with a duration of 10 seconds or longer.

## 2.3 Model-Based Data Purification

We further conduct a third strict filtering of the videos at the high-level model layer.

**Video aesthetic score.** The Video Training Suitability Score (VTSS) proposed in Koala-36M [36] integrates multiple pieces of manually labeled information regarding dynamic and static qualities, which enables a comprehensive evaluation of the quality of each video. We extract the native vtss score for each video (scaled within the range from -0.0575 to 0.0728) and filter out the data with a vtss score less than 0.01.

**Temporal motion score.** For model training, videos with subjects or camera movements that are either too slow (static frames lacking motion information) or too fast (unstable shots causing blurriness) are not ideal. Therefore, we use RAFT [33] to sample the motion relationships between temporal frames at intervals. After calculating the global average, we filter the data to retain values between 0.1 and 100.

**Video-caption consistency.** After obtaining the summarized caption of the video according to Sec. 2.4, we use VideoCLIP-XL-v2 [35] to get the similarity scores of all pairs, and filter the data with lower caption similarity by setting a threshold of 0.2.

**MLLM-assisted attribute judgment.** Before archiving the final data, we use Qwen2.5-VL-72B [2] to output binary judgments of low-quality attributes for each video. These attributes include 16 types such as Transition Effects, Watermarks, Split Screens, Screen Recordings, Picture-in-Picture, *etc*. If any of these low-quality attributes are detected, the corresponding video will be deleted.

Considering that we have already filtered out the low-quality data during the video collection process, and after the above statistical and model-based filtering procedures, the quality of the clips in the UltraVideo can be greatly ensured. Finally, we obtained 42K short videos in 3s~10s and 17K ≥10s long videos.

## 2.4 Comprehensive Structured Caption

Detailed captions are of great importance for fine-grained controllable video generation. Recent video datasets (*e.g*., Koala-36M [36]) and video generation methods (*e.g*., Wan2.1 [34] and Hunyuan-Video [15]) have demonstrated that detailed captions are essential for model training and application. Thus, long captions have become a key factor in the development of video datasets. Recent video generation models also primarily support long captions as input: for instance, Wan2.1/2.2 uses umt5-xxl [44] as the text encoder, while HunyuanVideo employs MLLMs for multimodal encoding. However, most current datasets focus more on the quantity of videos with simple captions. We fully utilize the capabilities of open-source foundation (M)LLMs to automatically construct comprehensive and high-quality structured captions.

**Structured description.** To achieve high-quality video generation, some recent datasets have attempted to generate structured captions to provide better text-video consistency. Typically, Mira-data [14] combines 8 evenly selected frames into a 2×4 image, and together with the "short" hint from Panda-70M, it is fed into GPT-4V to generate a "dense caption", and then, under carefully designed prompts, an additional 4 types of structured descriptions are obtained in a single dialogue turn. The recent Koala-36M [36] uses GPT-4V to generate structured video captions for fine-tuning the LLaVA caption model, which is used to generate captions containing 6 types of structured information with an average of 202.3 words per video. Different from the above solutions that use the closed-source GPT-4V, we propose a structured captioning solution based on the open-source Qwen2.5-VL-72B [2], which can be easily ported for local deployment and continuously enhance its capabilities as open-source community models are updated. Specifically, it includes 9 categories: 1) Brief Description. 2) Detailed Description. 3) Background. 4) Theme Description. 5) Style. 6) Shot Type. 7) Camera Movement. 8) Lighting. 9) Video Atmosphere. Fig. 4 and Fig. A4 show the distribution of each type of caption, from which it can be seen that our caption system is able to generate more fine-grained descriptions for text-to-video training.

Table 1: Comparison popular text-to-video datasets. Our UltraVideo is a *high-resolution* and *high-quality* premium T2V dataset, featuring *comprehensive structured captions* with a significantly longer average caption length. In addition to the main short version ranging from 3 seconds to 10 seconds, we also list the derived long version (*:) that exceeds 10 seconds for potential future research.

| Dataset | Resolution | Structured Caption | Average Caption Length | Average Video Length | Duration | Video Clips | Year |
|---|---|---|---|---|---|---|---|
| HowTo100M [19] | 240p | None | 4.0 words | 3.6s | 134.5Khr | 136M | 2019 |
| WebVid-10M [3] | 360p | None | 12.0 words | 17.5s | 52Khr | 10M | 2021 |
| HD-VILA-100M [43] | 720p | None | 32.5 words | 13.4s | 371.5Khr | 103M | 2022 |
| InternVid [38] | 720p | None | 17.6 words | 11.7s | 760.3Khr | 234M | 2023 |
| Panda-70M [8] | 720p | None | 13.2 words | 8.5s | 166.8Khr | 70.8M | 2024 |
| MiraData [14] | 720p | 6 | 318.0 words | 72.1s | 16Khr | 330K | 2024 |
| VIDGEN-1M [31] | 720p | None | 89.3 words | 10.6s | 2.9Khr | 1M | 2024 |
| LVD-2M [41] | 720p | None | 88.8 words | 20.2s | 14.6Khr | 2.1M | 2024 |
| Koala-36M [36] | 720p | 6 | 202.3 words | 13.6s | 137Khr | 36M | 2024 |
| OpenSoraPlan [16] | 1080p | None | 100.2 words | 20.1s | 2.8Khr | 512K | 2024 |
| OpenVid-1M [20] | 720p | None | 126.5 words | 7.2s | 2.1Khr | 1M | 2025 |
| OpenVidHD-0.4M [20] | 1080p | None | 104.5 words | 9.6s | 1.2Khr | 433K | 2025 |
| VideoUFO [37] | 720p | 2 | 155.5 words | 12.6s | 3.5Khr | 1M | 2025 |
| UltraVideo-Long (Ours)* | 4K / 8K | 10 | 850.3 words | 30.9s | 143hr | 17K | 2025 |
| UltraVideo (Ours) | 4K / 8K | 10 | 824.2 words | 5.3s | 62hr | 42K | 2025 |

**LLM-based caption summarization.** Different structured captions may potentially have different preferences due to variations in prompts during their construction. Therefore, based on the open-source Qwen3-4B [32], we integrate the above sub-captions to obtain a summarized description, which serves as one of the additional text prompt options.

## 2.5 Statistical Comparison and Analysis

**Comparison with popular video-text datasets.** Tab. 1 compares the properties of different popular T2V datasets. Our UltraVideo is the first to push T2V data to UHD-4K/-8K resolution and features more comprehensive structured captions for model fine-tuning. This dataset prioritizes higher visual quality over quantity, yet its volume of 42K samples still represents a substantial scale.

**Resolution *vs.* FPS.** UltraVideo provides the native video resolution and frame rate, potentially supporting future research such as video frame interpolation. Tab. 2 demonstrates the distribution.

**Numerical Statistics from Multiple Perspectives.** Fig. 4 displays the statistical information of UltraVideo from multiple perspectives to better help users achieve a more detailed understanding. (a) As described in Sec. 2.1, we confirmed seven major themes with diverse topics with the assistance of LLM. The upper-left corner shows a diverse distribution that can promote more generalizable T2V learning. (b) After strict screening in Sec. 2.3, each evaluation model scores at a high level, ensuring the high quality of the dataset. How-

Table 2: Resolution *vs.* FPS statistics.

| Type | #Reso. / FPS | $\leq 30$ | $\geq 50$ | All |
|---|---|---|---|---|
| | 4K | 24,749 | 7,978 | 32,727 |
| Short | 8K | 6,278 | 3,179 | 9,457 |
| | Sum | 31,027 | 11,157 | 42,184 |
| | 4K | 6,324 | 5,953 | 12,277 |
| Long | 8K | 1,822 | 2,498 | 4,320 |
| | Sum | 8,146 | 8,451 | 16,597 |

ever, users can still further filter based on these scores for stricter criteria. (c) The distribution of video duration and total frame count in short and long video sets. (d) The length distributions of typical "Brief Description", "Detailed Description", "Summarized Description", and the aggregated captions. Structured and detailed captions help improve the capability of fine-grained controllable video consistency. (e) An intuitive word cloud to visualize the captions.

**Analysis of non-compliance.** We selected the recent Koala-36M [36] for a video quality comparison. We randomly sampled 1000 videos each and had five different people evaluate them (we defaulted to using short videos). A video was considered a "bad video" if it had any of the following issues: Subtitles, Abnormal Color Patches, Green Screen, Blue Screen, Transition Effects, Watermarks, Stickers, Borders, Split Screens, Screen Recordings, Picture-in-Picture, Still Video, Blurred Video, Scrambled Video, and Solid-Color Backgrounds. Since the UltraVideo inherently has a high resolution above 4K and high image quality, we informed each subject to ignore this factor when making judgments about the results. In the end, the UltraVideo had a failure rate of 2.3%, which is significantly lower than the 41.5% failure rate of the popular Koala-36M. This proves the effectiveness of our curation process and implies that the UltraVideo is undoubtedly the current "quality champion" in the video community.

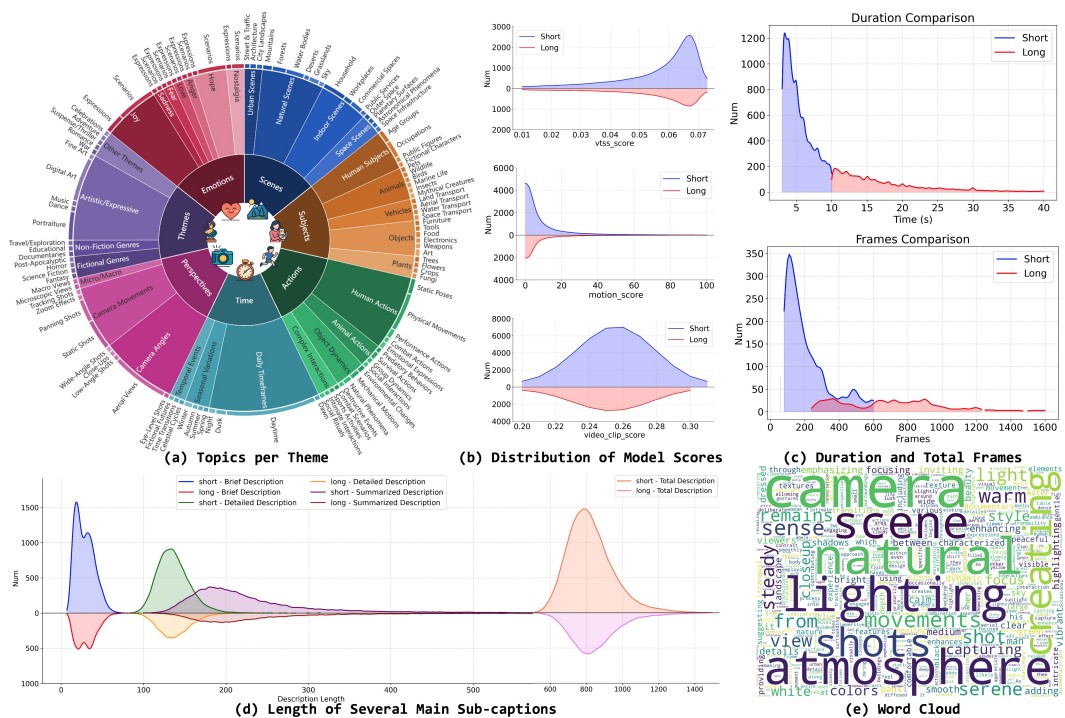

Figure 4: Statistical distributions of our **UltraVideo** from different perspectives.

## 3 UltraWan: Stand on the Shoulders of Giants

Based on the UltraVideo dataset, we explored the attempt of generating natively high-resolution videos, and specifically conducted fine-tuning experiments using Wan-T2V-1.3B [34] in this paper. We were surprised to find that just 42K exceptionally high-quality videos with comprehensive text are sufficient to have a significant impact on the aesthetics and resolution of the generated videos. Since we only use LoRA for fine-tuning without involving modifications to the model structure, the relevant experience can be easily transferred to other T2V models for the open-source community.

### 3.1 Resolution Scaling of Wan.

**Powerless extrapolation.** Benefiting from the relative position encoding and rotational invariance of RoPE, the DiT-based Wan has a certain degree of variable resolution inference capability. However, when we directly perform extrapolation on the native Wan-T2V-1.3B for 1K and 4K resolutions, we find that the performance deteriorates significantly or even becomes ineffective, as shown in Fig. 2. Therefore, the high-resolution inference capability requires model parameters that are adaptable, which has triggered our exploration of scaling the Wan model.

**Structural configures for UltraWan-1K and UltraWan-4K.** For high-resolution T2V generation, the memory calculation amount of the model will increase significantly. Therefore, we use the smaller

Table 3: Model configures.

| Model / CFG | H×W×T | B.S. | Train/Infer Mem. | GPU Hours |
|---|---|---|---|---|
| UltraVideo-1K | 1088×1920×81 | 128 | 58.5G/18.4G | 3.4K |
| UltraVideo-4K | 2160×3840×29 | 128 | 83.7G/25.7G | 7.6K |

Wan-T2V-1.3B to conduct experiments with H20 GPUs. Specifically, our UltraWan-1K maintains an output of 81 frames, while UltraWan-4K reduces the number of output frames to 29 to ensure that a single sample can fit on a single GPU card. Tensor parallel is not used and the batch size per GPU is 1, and the GPU memory usage during training and inference is shown in Tab. 3.

### 3.2 Training Scheme.

**Random caption sampling strategy.** To make full use of comprehensive structured captions for fine-grained prompt control, we propose a random caption sampling strategy. Specifically, with a probability of 1/3, we select from i) Brief Description, ii) Detailed Description, and iii) Summarized

Table 4: VBench evaluation results per dimension. *: Videos are downsampled to 1K to avoid OOM.

| Models | Subject Consistency | Background Consistency | Temporal Flickering | Motion Smoothness | Dynamic Degree | Aesthetic Quality | Imaging Quality | Object Class |
|---|---|---|---|---|---|---|---|---|
| Wan-T2V-1.3B-480p [34] | 96.11% | 98.06% | 99.09% | 98.75% | 27.77% | 65.83% | 68.91% | 66.66% |
| Wan-T2V-1.3B-1K [34] | 95.86% | 98.15% | 98.07% | 98.75% | 66.66% | 54.82% | 55.12% | 33.33% |
| UltraWan-1K (Full) | 95.71% | 97.94% | 98.86% | 99.06% | 22.22% | 61.52% | 67.39% | 66.66% |
| UltraWan-1K (LoRA) | 97.27% | 98.26% | 99.33% | 98.62% | 66.66% | 62.5% | 67.74% | 82.29% |
| UltraWan-4K (LoRA) | 96.05% | 98.02% | 98.88% | 98.47%* | 66.66%* | 56.81% | 71.61% | 50.00% |

| Models | Multiple Objects | Human Action | Color | Spatial Relationship | Scene | Appearance Style | Temporal Style | Overall Consistency |
|---|---|---|---|---|---|---|---|---|
| Wan-T2V-1.3B-480p [34] | 51.04% | 66.66% | 100.0% | 100.0% | 08.33% | 20.54% | 24.39% | 25.31% |
| Wan-T2V-1.3B-1K [34] | 25.00% | 22.22% | 100.0% | 36.66% | 00.00% | 18.75% | 12.24% | 20.65% |
| UltraWan-1K (Full) | 47.91% | 66.66% | 100.0% | 50.00% | 16.66% | 17.85% | 19.81% | 24.27% |
| UltraWan-1K (LoRA) | 49.58% | 66.66% | 100.0% | 75.76% | 18.22% | 19.57% | 23.34% | 23.99% |
| UltraWan-4K (LoRA) | 42.75% | 66.66% | 100.0% | 100.0% | 00.00% | 19.46% | 19.31% | 22.88% |

Description. If either the Brief Description or the Detailed Description is sampled, we then randomly select one caption from the remaining 7 categories mentioned in Sec. 2.4 for supplementation, which serves as the final prompt fed into the model.

**Sub-clip sampling.** For each video, we uniformly sample an average number of frames from the middle to both sides according to the number of training frames to ensure the consistency between the sub-clip and the caption. In the experiment, we keep the native FPS of the video and perform sampling without intervals.

**Memory-efficient HDR plugins of Wan-1K/-4K LoRA.** Considering the computational power and memory requirements for fine-tuning, we use LoRA for parameter-efficient fine-tuning. The rank is set to 64/16 for UltraWan-1K/UltraWan-4K, and the modules affected are QKV in the self-attention and the output linear layer, as well as the first and third linear layers in the feedforward network.

**Hyperparameter setting.** We use AdamW [17] with betas=(0.9, 0.999), weight_decay=1e-2, and learning_rate=1e-4. Both UltraWan-1K and UltraWan-4K are trained for one epoch.

## 4  Experiments

Limited by the significant increase in computational power and video memory caused by high resolution, this paper only conducts experiments on the small-scale Wan-T2V-1.3B [34] to: *1)* propose and implement the training of native 1K/4K T2V models for the first time; *2)* demonstrate the high-quality effectiveness of the dataset.

**Comparison results for high-resolution video generation.** Limited by the slower inference caused by increased computational power for high resolution, we randomly sample one-tenth (≃96) of the prompts from VBench [13] for testing. As shown in Tab. 4, we compare five models: *i)* official Wan-T2V-1.3B with 480×832 resolution. *ii)* increasing the resolution to 1K (1088×1920). *iii)* 1K full finetuning. *iv)* 1K LoRA PEFT. *v)* 4K LoRA PEFT. The following conclusions can be drawn from the results: *1)* Scaling the official model to 1K leads to a significant decline in performance. *2)* The full-parameters training based on UltraWan-1K has significantly improved generation at 1K resolution, but differences in training hyperparameters (such as batch size and prompts) from the native model may cause its results to be slightly worse overall than the LoRA model based on UltraWan-1K. Considering training costs, we recommend using the LoRA-based UltraWan-1K scheme. *3)* The higher UltraWan-4K model performs better in indicators related to image quality and temporal stability, but its lower frame rate (inference uses 33 frames to ensure the time exceeds 1s) causes some indicators to be worse compared to UltraWan-1K. Fig. 5 shows the qualitative effect comparison. The official Wan-T2V-1.3B cannot directly generate high-resolution 1K videos, while our UltraWan is capable of handling semantically consistent 1K/4K generation tasks.

**Inaccurate metrics for high-resolution video evaluation.** Some existing metrics are not suitable for evaluating high-resolution videos and need improvement. For example, metrics such as Background Consistency and Dynamic Degree may produce conclusions contrary to human visual perception. Metrics like Human Action and Color are less discriminative due to being affected by model accuracy

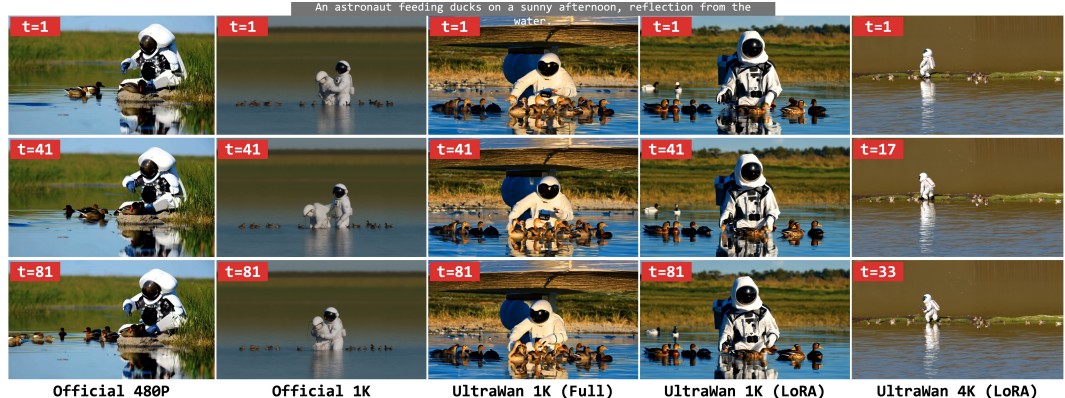

Figure 5: Intuitive results with the prompt in VBench [13]. Enlarged for better visual effects.

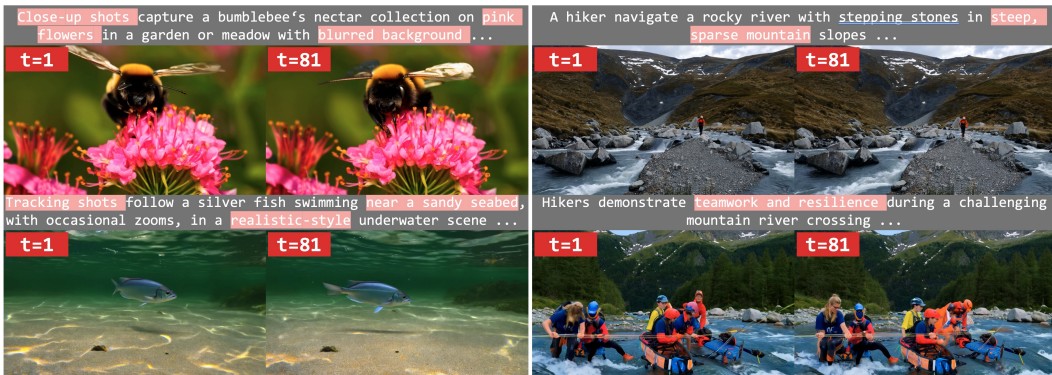

Figure 6: Our UltraWan-1K is capable of generating semantically consistent videos.

and task difficulty. Direct evaluation of 4K videos using Motion Smoothness and Dynamic Degree can cause out-of-memory (OOM) issues, urgently requiring replacement with more modern models.

**Semantic consistency with fine-grained captions.** Thanks to the structured captions in UltraVideo during training, our UltraWan exhibits stronger semantic consistency, as shown in Fig. 6.

**Human study for 1K video generation.** To demonstrate the effectiveness of the proposed UltraWan, we conducted a human preference experiment. Specifically, we used the videos generated from the aforementioned VBench test subset as test samples and built a visual interface using streamlit [30] to ask 10 subjects about their preferences across three dimensions: video quality aesthetics, temporal stability, and text consistency. As shown in Fig. 2, the official Wan-T2V-1.3B struggles

Table 5: Human preferences.

| Metric | Official Wan | UltraWan |
|---|---|---|
| video quality aesthetics | 18.90% | 81.10% |
| temporal stability | 44.30% | 55.70% |
| text consistency | 45.50% | 54.50% |

to maintain content quality when generating 1K videos, so it retains the officially recommended 480×832 resolution output. As shown in Tab. 5, thanks to the high-resolution fine-tuning on the high-quality UltraVideo, UltraWan-1K has a significant advantage in video quality aesthetics, while showing similar tendencies in temporal stability and text consistency.

**Ablation study on filtered and unfiltered data.** The unfiltered 1K video dataset (that is, one that does not include Statistical Data Filtering in Sec. 2.2 and Model-Based Data Purification in Sec. 2.3 contains 62K short videos. We randomly selected a 10K subset from it for LoRA fine-tuning at 1K resolution for one epoch. Additionally, we also randomly selected 10K subsets from OpenVidHD-0.4M and OpenSoraPlan for this purpose, labeled as UltraWan-1K-unfiltered-10K, UltraWan-1K-OpenVidHD-10K, and UltraWan-1K-OpenSoraPlan-10K, respectively. Results of the VBench subset are compared in Tab. 6. Thanks to strict control over video sources and secondary manual preview, the quality of the unfiltered sub-dataset is slightly inferior to that of the final filtered dataset (UltraWan-1K-

Table 6: Ablation on different sub-datasets.

| Model | Subject Consistency | Background Consistency | Aesthetic Quality | Imaging Quality |
|---|---|---|---|---|
| UltraWan-1K-42K | 97.27% | 98.26% | 62.50% | 67.74% |
| UltraWan-1K-10K | 97.05% | 98.25% | 62.28% | 67.53% |
| UltraWan-1K-unfiltered-10K | 96.86% | 98.19% | 62.12% | 67.17% |
| UltraWan-1K-OpenVidHD-10K | 96.71% | 98.07% | 60.25% | 64.92% |
| UltraWan-1K-OpenSoraPlan-10K | 96.84% | 98.15% | 61.58% | 66.15% |

unfiltered-10K *v.s.* UltraWan-1K-10K). However, UltraWan-1K-unfiltered-10K still shows significant advantages over UltraWan-1K-OpenVid-10K and UltraWan-1K-OpenSoraPlan-10K, especially in terms of improved image quality. In addition, the metrics of UltraWan-1K-10K also slightly decrease compared to the full-scale model UltraWan-1K-42K. Through visual comparative analysis of the results, we found that UltraWan-1K-unfiltered-10K has slightly weaker semantic adherence capability compared to the filtered version in some scenarios—especially for subjects involving motion, which are more prone to artifacts. Nevertheless, it still outperforms the results trained by OpenVidHD-10K and OpenSoraPlan-10K, which validates the effectiveness of our curation pipeline.

## 5 Conclusion

The quality of video datasets, including image quality, resolution, and fine-grained captions, is a critical determinant of the performance ceiling for video generation models. The escalating demands of video applications, particularly for UHD-4K content, highlight the inadequacy of existing public datasets. In response, we introduced UltraVideo, a high-quality open-source UHD-4K/8K text-to-video dataset that encompasses diverse topics and provides comprehensive structured captions for each video. Our innovative four-stage automated curation process ensures data excellence, addressing key challenges in resolution scalability and semantic granularity. By extending the Wan model to UltraWan-1K/-4K, we demonstrated enhanced capabilities in natively generating high-resolution videos with superior text controllability. This work not only fills a significant gap in high-resolution video generation research but also advances the state-of-the-art through novel dataset construction, advanced data processing pipelines, and refined model architectures, paving the way for future breakthroughs in UHD video generation.

**Limitations, broader impact and social impact.** Thanks to the preservation of native resolution, frame rate, and audio in UltraVideo, it can be readily adapted to any relevant video tasks in ultra-resolution settings, such as exploring low-level UHD video super-resolution/frame interpolation/codecs, and high-level video editing/frame-to-frame/music generation. Additionally, we plan to leverage the long-duration subset for in-depth exploration of long-form video generation tasks in the future. The proliferation of fake videos may trigger the spread of false information, and the malicious use of AI-generated content will seriously threaten information authenticity and social stability. There is an urgent need to establish multi-dimensional regulatory frameworks and technical response solutions.

**Acknowledgments and Disclosure of Funding.** This work was supported by a Grant from The National Natural Science Foundation of China (No. 62525309)

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

# Appendix

## Overview

The appendix presents the following sections to strengthen the main manuscript:

— We provide a demo video, and source code in the Supplementary Materials for a better understanding of our work. Additionally, we offer the UltraVideo dataset at https://huggingface.co/datasets/APRIL-AIGC/UltraVideo and the UltraWan model weights at https://huggingface.co/APRIL-AIGC/UltraWan.

— **Sec.** A shows the Related Work part of the paper.

— **Sec.** B presents some experimental findings of UltraWan.

— **Sec.** C shows more statistical distributions of comprehensive 10 structured captions.

— **Sec.** D shows more qualitative results

## A    Related Work

### A.1    Text-to-Video Datasets

The release of the early LAION series datasets [27, 28] has facilitated the emergence of subsequent high-quality text-to-image foundation models represented by SD/SDXL [25, 23] and others. With the booming popularity of SORA [21], the research on text-to-video has gained more momentum, and there is a more pressing need for relevant datasets. Early researchers have constructed a large number of video-text datasets for specific scenario tasks. For example, UCF101 [29] is for action recognition, and MSVD [6] and MSR-VTT [42] are for video retrieval. *However, most of these datasets adopt manual annotation, which requires higher costs, leading to limitations in scale. Moreover, the quality of early videos is poor, and the annotations are not suitable for modern video generation tasks.* In order to alleviate the above problems, WebVid-10M [3] has collected 10.7 million general videos with alt-text. However, its videos contain low-quality watermarks. Meanwhile, works [19, 43, 46] have proposed using ASR to automatically annotate videos. Recently, InternVid [38] has constructed a video-centric multimodal dataset. Panda-70M [8] has become the largest publicly available video dataset, but it contains too many low-quality videos with simplistic and incomplete descriptions. VidGen-1M [31], on the other hand, has screened high-resolution and long-duration videos from the HD-VILA data through a coarse-to-fine curation strategy. MiraData [14] focuses on long-duration videos with detailed and structured captions. Koala-36M contends that temporal splitting, detailed captions, and video quality filtering determine the quality of the dataset. It contains 36 million high-quality video-text pairs. While LVD-2M [41] includes long-take videos with significant motion and temporally-dense captions. The recent OpenVid-1M [20] provides a precise high-quality dataset with expressive captions. However, most of the latest datasets only offer 720p videos. Only a few methods provide 1080p data, such as OpenVidHD-0.4M [20] and OpenSoraPlan [16]. There is still no publicly available dataset with a resolution of 4K and above to meet the growing application demands of high-resolution video generation. We supplement existing video datasets with high-quality 4K/8K videos featuring comprehensive captions to support Ultra-High-Definition video-centric generation.

### A.2    Video Generation Models

Thanks to the progress of deep learning architectures and the emergence of large-scale datasets, video generation models have made remarkable progress in recent years. From the early Generative Adversarial Networks (GAN) [10] to the diffusion models [12] in recent years, there have been qualitative improvements in the quality, diversity, and controllability of generated videos. Diffusion first achieved good results in the field of images [25, 25, 23], and then extended the temporal dimension to the field of video generation represented by AnimateDiff [11] and SVD [4]. With Sora triggering the application of commercial video models, a series of text-to-video models have emerged one after another [40, 7], and the architecture has also transitioned from the early UNet-based architecture to the DiT-based architecture [45, 16, 18]. Motivated by the large language model (LLM) field, some autoregressive-based solutions have also been proposed [39, 9, 26]. Currently, the most popular open-source models, HunyuanVideo [15] and Wan [34], have attracted much attention due

to their good generation quality. This paper for the first time explores the native 4K text-to-video generation based on Wan.

## A.3 Video Data Curation

Data curation is particularly important for the quality of large-scale video datasets. The prevailing process relies on image models for tagging and manual rule-based curation. For example, CLIP [24] measures the consistency between images and texts, and LAION-Aesthetics [1] evaluates the aesthetics of images. These metrics are usually averaged over time to serve as video metrics without taking into account the temporal characteristics of videos. SVD [4] provides a comprehensive overview of the management process of large-scale video datasets, including techniques such as video clipping, captioning, and filtering. Some subsequent works have recognized the importance of data curation. VidGen-1M [31] has designed a three-stage process of coarse curation (scene splitting, tagging, and sampling), captioning, and fine curation with a large language model (LLM). Koala-36M [36] proposes a refined data processing pipeline, including transition detection methods, a structured caption system, and the Video Training Suitability Score (VTSS) for data filtering. LVD-2M [41] creates an automatic pipeline for video filtering and long video recaptioning. The quality of video datasets greatly affects the upper limit of the performance of video generation models, and the data curation process determines the quality of videos and captions. We have carefully designed a curation process based on multiple modern foundation models to obtain high-quality videos and structured comprehensive captions.

## B    Experimental Findings of UltraWan

Considering the high computational cost of 1K/4K video generation, we have provided intermediate results at https://huggingface.co/datasets/APRIL-AIGC/UltraVideo for research reference. We conducted comparative analysis of Wan-T2V-1.3B (480P for optimal performance), UltraWan-1K, and UltraWan-4K (see Fig. A1-Fig. A3), with key findings: *1)* As a LoRA-tuned model, UltraWan's capabilities are primarily constrained by: (i) the base Wan-T2V-1.3B model's capacity, (ii) tuning dataset quality, and (iii) optimization strategy. Quantitative results in Table 4 demonstrate UltraWan-1K's competitive performance, attributable to UltraVideo's high-quality training data. *2)* Given identical prompts, all three models generate similar scenes. However, UltraWan-1K/4K exhibit superior semantic alignment compared to Wan-T2V-1.3B while achieving native 1K generation. *3)* UltraWan-4K shows increased artifact susceptibility for subjects versus UltraWan-1K, likely due to: (i) significant resolution gap complicating LoRA adaptation, and (ii) potential undertraining (limited to 1 epoch due to compute constraints). Notably, it demonstrates preference for landscapes and architectural scenes. *4)* Shared architecture leads to consistent artifacts in challenging scenarios across all models (*e.g.*, Fig. A3). *5)* VRAM limitations restrict UltraWan-4K to 29-frame training (vs. base model's 81 frames), increasing learning difficulty. Future work should investigate memory-efficient video foundation models.

## C    Statistical Distributions of Comprehensive 10 Structured Captions

Fig. A4 shows more statistical distributions of Comprehensive 10 structured captions.

## D    More Qualitative Results

Fig. A5 shows more qualitative results.

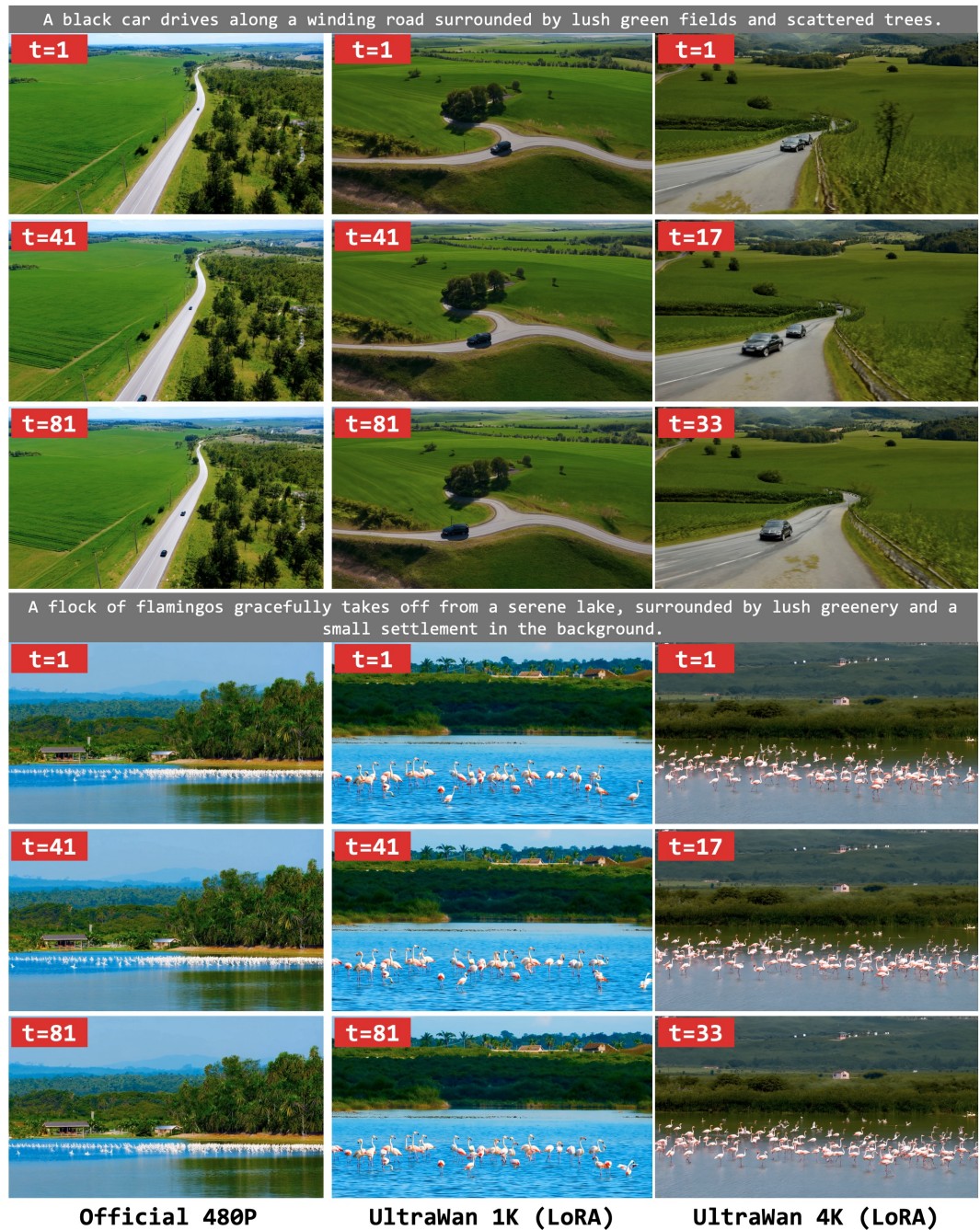

Figure A1: Intuitive results with the prompt in UltraVideo. Enlarged for better visual effects.

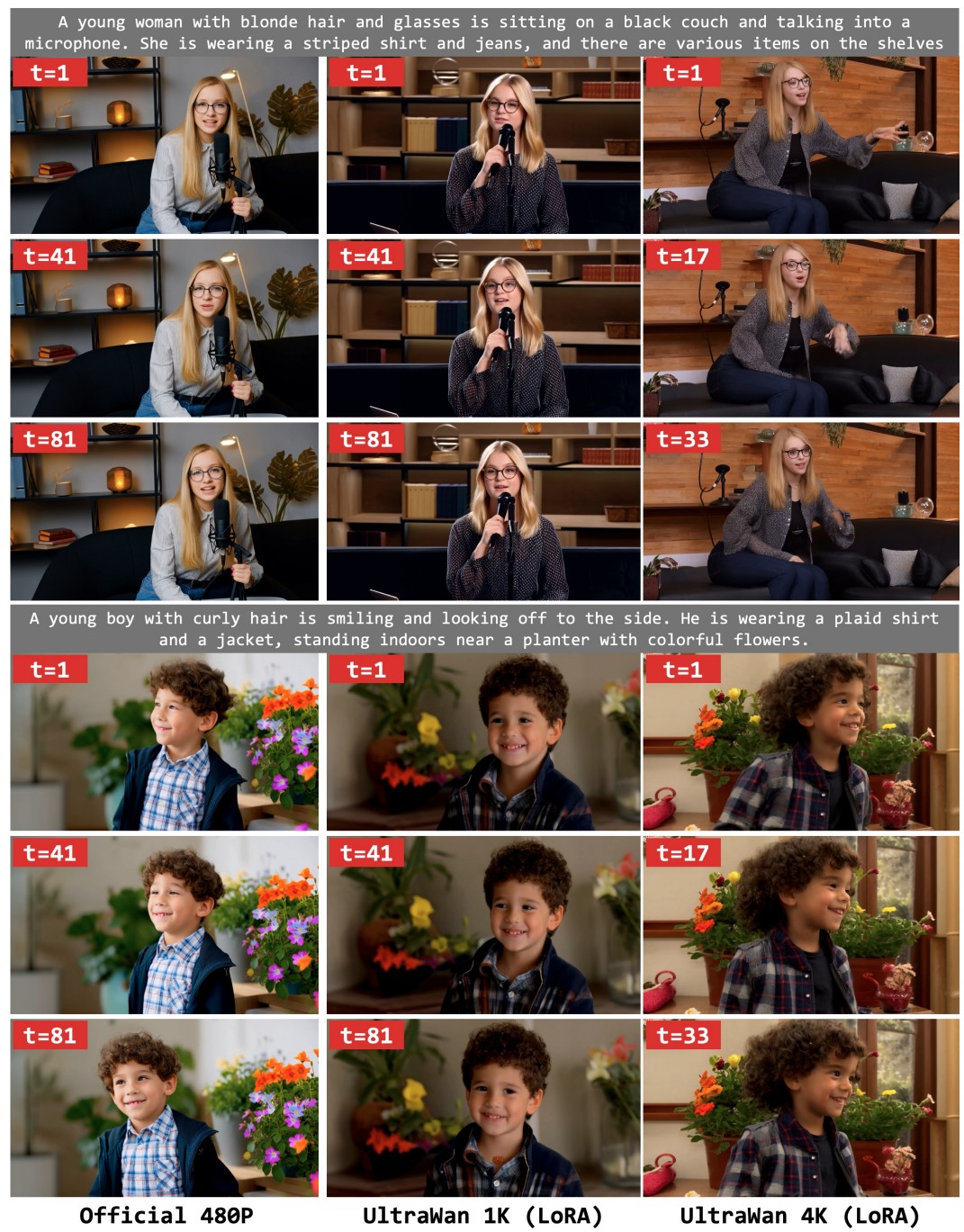

Figure A2: Intuitive results with the prompt in UltraVideo. Enlarged for better visual effects.

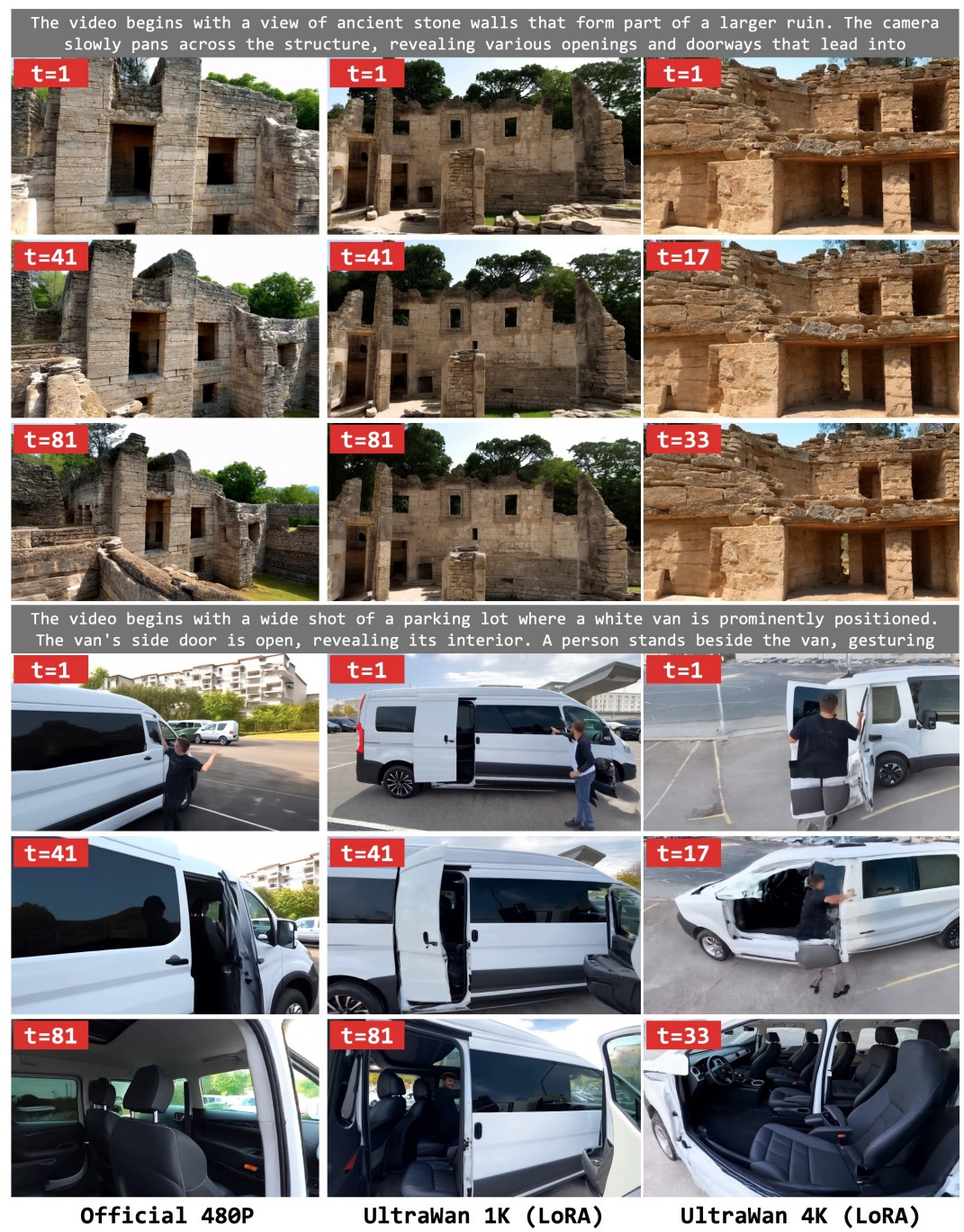

Figure A3: Intuitive results with the prompt in UltraVideo. Enlarged for better visual effects.

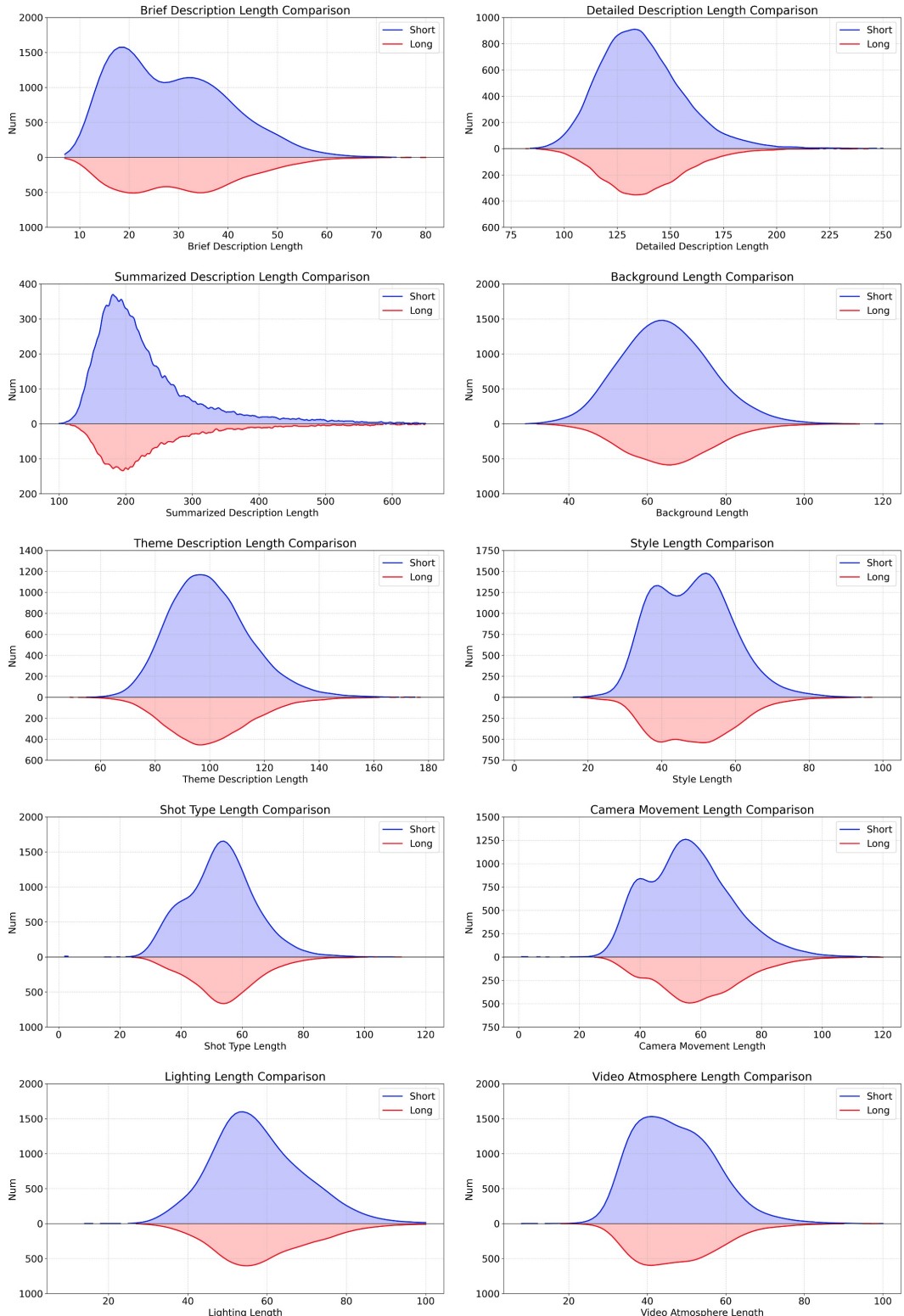

Figure A4: Statistical distributions of comprehensive 10 structured captions.

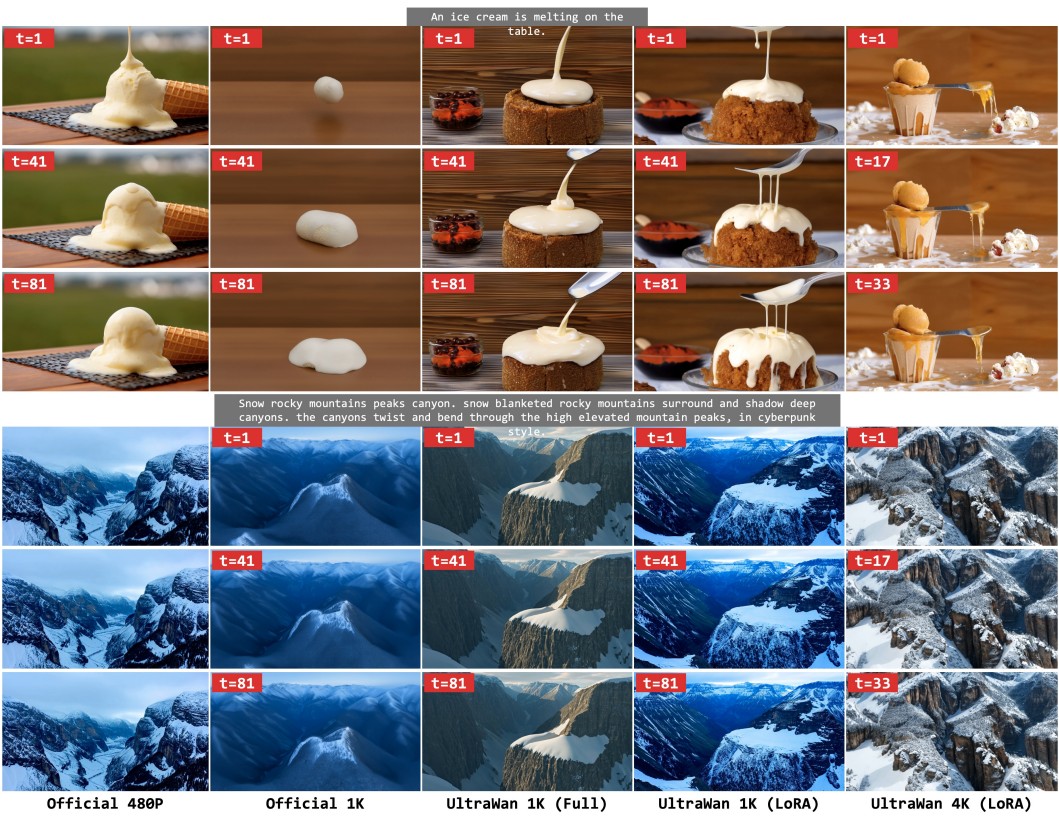

Figure A5: Intuitive results with the prompt in VBench [13]. Enlarged for better visual effects.

