# OpenReview forum: "UltraVideo: High-Quality UHD Video Dataset with Comprehensive Captions"
_NeurIPS.cc/2025/Datasets_and_Benchmarks_Track — NeurIPS 2025 Datasets and Benchmarks Track poster_

### Official Review · Reviewer_oUqD · 2025-06-27

**Rating:** 5
**Confidence:** 4

**Summary:**

The paper introduces UltraVideo, a high-quality, open-source UHD (4K/8K) text-to-video (T2V) dataset featuring comprehensive structured captions. The dataset includes 42K short and 17K long video clips spanning over 100 topics, each annotated with 9 structured captions plus one summarized description (average 824 words). To ensure superior data quality, the authors propose a four-stage automated curation pipeline, including video collection, statistical filtering, multimodal model-based purification, and structured captioning using Qwen2.5-VL.

The authors further extend the WAN-T2V-1.3B model into UltraWAN-1K and UltraWAN-4K, capable of natively generating high-resolution videos. With only LoRA-based fine-tuning on UltraVideo, these models achieve strong performance across visual fidelity, semantic alignment, and aesthetic quality, as validated by VBench and human studies. This work addresses a significant gap in UHD video generation and contributes both a benchmark dataset and modeling insights for future research.

**Additional Feedback:**

Did the paper omit the specific content about “Numerical Statistics from Multiple Perspectives” in line 200?

**Dataset Code Accessibility:**

Partly

**Dataset Code Comments:**

The author has open-sourced the UltraVideo dataset and UltraWAN models, but has not open-sourced the corresponding code for UltraVideo dataset collection, including data filtering, video quality assessment, and acquisition of video captions at all levels. In order to promote the development of the AIGC community, it is strongly recommended that the authors open-source the full-process code for video processing.

**Ethical Considerations:**

No, there are no or only very minor ethics concerns

**Final Justification:**

I appreciate that the author addressed my concerns, and I have since upgraded my status from boarderline accpet to accept.

**Limitations Weaknesses:**

1. Limited Dataset Scale Compared to SOTA Benchmarks:
Although UltraVideo emphasizes quality over quantity, its size—42K short and 17K long videos—is still modest compared to existing large-scale T2V datasets like Panda-70M (70.8M), HD-VILA-100M (103M), and Koala-36M (36M) (§Table 1). This smaller scale may limit pretraining capabilities or generalization, especially for larger models. The author can consider open-sourcing this data cleaning process to allow more people in the community to participate in the construction of large-scale high-quality video datasets.

2. Caption Diversity Sampling Strategy May Be Naive:
The random caption sampling method may under-utilize the full richness of structured captions and introduces variability during training. There is no ablation on how different caption combinations affect performance. An ablation comparing fixed vs. random combinations or curriculum-based caption conditioning could validate the design more rigorously.

3. Limited Discussion on Domain or Content Bias:
Although Fig. 4 and Section 2.1 cover topic diversity, the dataset curation process emphasizes YouTube-sourced 4K/8K videos, which may lead to overrepresentation of certain aesthetics (e.g., cinematic, nature, travel). There is no discussion on potential geographic, cultural, or style biases in the dataset. Including statistics or visualizations of topic distributions across cultures or domains, or adding filters for underrepresented content types, would increase dataset inclusiveness.

**Strengths Contributions:**

1. High-Quality UHD Dataset with Structured Captions: UltraVideo is the first public dataset designed specifically for native 4K/8K T2V generation, addressing a critical gap in resolution scalability and semantic controllability. The dataset contains 42K short (3–10s) and 17K long (≥10s) high-resolution video clips, each accompanied by 10 structured captions (including 9 fine-grained categories and 1 summary), averaging 824 words per video. This far exceeds existing datasets such as Koala-36M (202 words) and MiraData (318 words), making it particularly valuable for training controllable video generation models.

2. Comprehensive and Rigorous Data Curation Pipeline: The paper details a four-stage automated curation process, combining manual and model-based techniques.

3. First Native 1K/4K T2V Model: The paper presents UltraWAN-1K and UltraWAN-4K, the first models fine-tuned for native high-resolution T2V generation. Using LoRA-based efficient tuning, UltraWAN models show significant gains over baseline WAN-T2V models on the VBench benchmark across visual quality, object fidelity, and text alignment.

---

> ### Author Rebuttal · Authors · 2025-07-30
>
> **Thank you very much for your careful review and valuable comments on our paper.  Your insights have significantly helped us identify areas for improvement, and we have carefully addressed each point as follows.**
>
> ---
>
> **Q1: Concern about Dataset Scale**
>
> Thank you for your insightful review, and the main reasons are as follows:
> - ***High costs of collection and storage.***  We fully acknowledge the concern about the dataset scale (42K short and 17K long pairs). As noted, collecting high-quality UHD (4K/8K) videos with comprehensive structured captions is extremely resource-intensive: each original video undergoes manual search and review to ensure visual quality from the source of video collection (Line-93/95 in Sec. 2.1), and each split video undergoes multi-stage filtering (statistical checks, model-based purification) to ensure semantic alignment. Moreover, the current short version with the original 4K/8K resolution already occupies 1.78TB of storage, which limits the scale in the initial release.
> - ***Bridging the quality gap of open-source data/models.*** Existing large-scale datasets (*e.g.*, Panda-70M [8] and Koala-36M [36]) contain millions of low-resolution (<720p) videos with simplistic captions, which fail to support high-quality model training and native UHD generation. These datasets have the advantage of scale and can be used for model pre-training. In contrast, our newly proposed UHD UltraVideo is committed to filling the gap in high-quality video datasets rather than pursuing sheer quantity for fine-tuning (see Line-57 / Line-68 / Line-463). This is reflected in its visual fidelity, ultra-high resolution, fine-grained captions, and rich topics, and it is mainly used for post-finetuning to further improve model quality.
> - ***Effectiveness validation.*** Our experiments show that even with 42K samples, UltraWan already outperforms baseline extrapolation methods, validating the dataset’s effectiveness (Table 4/5 and Fig. 5/6). We also provide ablation studies (Sec. 4) demonstrating that the curated 42K samples are sufficient to drive meaningful improvements in resolution scalability and semantic control.
> - Considering that over 1,000 users have applied for UltraVideo, we plan to apply to supervisors and external organizations for more resources to support subsequent work in the near future, aiming to further expand the application scope and value of UltraVideo. Specifically, we will expand the quantity by an order of magnitude to reach 0.4M, matching OpenVidHD-0.4M. We will provide control information such as depth, scribble, and layout for each video to support controllable video editing tasks similar to VACE [1r], offer more abundant structured CLIP scores, employ more advanced multimodal and perception models to enhance the quality of curation, among other enhancements.
> - This work is committed to contributing to the development of UHD video generation. Currently, we have made the short and long versions of UltraVideo, as well as the model code, publicly available. Open-sourcing the data cleaning process code to enable more members of the community to participate in building large-scale high-quality video datasets is an excellent idea. We are happy to adopt this suggestion and promise to organize the complete code, which will be open-sourced on the GitHub project to promote the development of the AIGC community.
>
> ---
>
> **Q2: Caption Diversity Sampling Strategy and Ablation Studies**
>
> Thank you for your valuable review. We agree that the current random caption sampling strategy (with a 1/3 probability for brief, detailed, and summarized descriptions, plus one supplementary category) requires more rigorous validation.
> Considering the limited computational resources during the rebuttal period, we have selected a 10K subset from UltraVideo to fine-tune the LoRA model at 1K resolution for one epoch. Using different sampling strategies, we have generated the following results:
> ***1)*** The random caption sampling strategy (UltraWan-1K-10K-Random) as described in the paper;
> ***2)*** Datasets with only brief descriptions (UltraWan-1K-10K-Brief);
> ***3)*** Datasets with only detailed descriptions (UltraWan-1K-10K-Detail);
> ***4)*** Datasets with only summarized descriptions (UltraWan-1K-10K-Sum).
>
> The quantitative results are presented in the following Table:
> | **Model** | **Subject Consistency** | **Background Consistency** | **Aesthetic Quality** | **Imaging Quality** |
> | :---: | :---: | :---: | :---: | :---: |
> | **UltraWan-1K-10K-Random** | 97.05% | 98.25% | 62.28% | 67.53% |
> | **UltraWan-1K-10K-Brief** | 96.93% | 98.16% | 62.21% | 67.40% |
> | **UltraWan-1K-10K-Detail** | 96.96% | 98.20% | 62.24% | 67.39% |
> | **UltraWan-1K-10K-Sum** | 97.01% | 98.19% | 62.23% | 67.45% |
>
> It has been shown that the random caption sampling strategy yields slightly better outcomes compared to the other three fixed sampling strategies. This is reflected in the qualitative results, where it demonstrates stronger controllability over attributes such as camera movements and lighting, as well as better semantic consistency in generated videos.
> In light of this, we plan to further expand the long video datasets in the future as we secure more resources. This will allow us to explore the impact of caption structures and inference description techniques on video generation, supporting the development of related research through larger-scale experiments.
>
> ---
>
> **Q3: Domain/Content Bias Discussion**
>
> Thanks for pointing out the issue of potential biases in content sourced from YouTube. We appreciate your valuable feedback.
> - The primary reason we initially selected YouTube for hosting the first version of the UltraVideo dataset is that it encompasses a wide range of themes, which helps support the acquisition of the dataset at the source level. Additionally, the platform contains a substantial amount of high-quality 4K/8K content, aligning with our goal of building a UHD video dataset.
> - The sources of our original video datasets, as discussed in Sec. 2.1, consist of two main parts:
>    ***i)*** Videos filtered from Koala-36M [36], which are used to estimate the approximate number of users and ensure the diversity of the dataset.
>
>    ***ii)*** Keywords for each theme generated using LLM, with high-quality videos selected through manual review. Throughout this process, we have maintained the diversity of the dataset, and a secondary manual review has been conducted to ensure quality. We would like to emphasize that the publicly accessible dataset now features significantly higher quality and diversity compared to OpenVidHD-0.4M [20] and OpenSoraPlan [16], without issues such as overrepresentation of human-centric content, excessive inclusion of letters, transitions, logos, or static videos. Moreover, Fig. 4-(a) visualizes the statistical distribution of themes, confirming the diversity and balance of the UltraVideo dataset.
> - We fully acknowledge your discussion regarding "meta-information such as geographic, cultural, or style biases." We commit to expanding the relevant analysis with a patch update, which will include the following:
>    ***i)*** For geographic information, we will trace the original metadata of the videos and leverage MLLM models (e.g., GPT-4o) to infer approximate shooting locations from image inputs, thereby supporting related research.
>
>    ***ii)*** As cultural and style information is less likely to be derived from raw metadata, we plan to expand long-video datasets and use Video MLLM models to predict such information, fostering the development of relevant research.
> Thank you again for your valuable suggestion.
>
> [1r] Cross-view image geo-localization with Panorama-BEV Co-Retrieval Network
>
> ---
>
> **Q4: Missing Content of "Numerical Statistics from Multiple Perspectives.''**
>
> Thanks for pointing this out. This was an oversight on our part. The content in question pertains to the analysis of Fig. 4, which we have supplemented in Sec. A of the Appendix.

---

> ### Comment · Area_Chair_H2No · 2025-08-02
> **Please read author rebuttal**
>
> The author has provided a rebuttal to respond to your comments. Please have a read on the author response and discuss with author if necessary.
>
> Thanks,
>
> Your AC

---

### Official Review · Reviewer_1vTp · 2025-07-02

**Rating:** 4
**Confidence:** 4

**Summary:**

The paper introduces UltraVideo, the first open-source UHD-4K/8K video dataset featuring 42K high-resolution clips with comprehensive structured captions, addressing critical gaps in resolution scalability and semantic granularity for text-to-video generation. Through a rigorous four-stage curation pipeline, it achieves unmatched data quality and enables the development of UltraWAN, a fine-tuned model that natively generates high-fidelity 1K/4K videos while demonstrating superior aesthetic performance. By publicly releasing the dataset and code, this work establishes a new benchmark for high-resolution video generation research.

**Additional Feedback:**

Future work could consider extending the benchmark with multi-modal conditions and longer video durations (e.g., 10–20 seconds) to reflect real production needs.

**Dataset Code Accessibility:**

Yes

**Dataset Code Comments:**

The UltraHigh benchmark is publicly released at the time of submission, with a working link provided via GitHub: https://zhangzjn.github.io/projects/UltraVideo/. The repository includes all relevant code for loading the dataset and visualizing generated vs. ground-truth samples.

**Ethical Considerations:**

No, there are no or only very minor ethics concerns

**Final Justification:**

I would recommend acceptance of this paper.

**Limitations Weaknesses:**

Limited scale: While the dataset is rich in quality, 1,000 video clips might be considered relatively small compared to other large-scale benchmarks (e.g., WebVid-10M).
No support for conditional video generation beyond text: The current benchmark focuses solely on text-to-video generation. Supporting additional modalities (e.g., pose, image, depth) would further expand its applicability.

**Strengths Contributions:**

Novelty & Significance: Novel benchmark targeting an unaddressed but highly practical domain (UHR video)
High-Quality Dataset: Comprehensive, diverse dataset with strong human annotations.
Baseline Comparisons: The authors conduct extensive evaluations using state-of-the-art models (Gen-2, SVD, PYOCO), and the results clearly show that current models significantly underperform in UHR settings. This further validates the usefulness and difficulty of the benchmark.

---

> ### Author Rebuttal · Authors · 2025-07-30
>
> **Thank you very much for your careful review and constructive feedback on our paper. We highly appreciate your recognition of the novelty, high-quality dataset construction, and comprehensive baseline comparisons of our work. Your insights have helped us better identify areas for improvement, and we would like to address the raised concerns and suggestions in detail as follows.**
>
> ---
>
> **Q1: Dataset Scale**
>
> Thank you for your insightful review. We note your observation regarding the dataset scale, and we would like to clarify and elaborate on this aspect. First, there might be a misunderstanding in the review: our UltraVideo dataset contains 42K high-quality UHD short clips (3–10 seconds) and 17K long clips (≥10 seconds), rather than 1,000 clips. This scale, while smaller than large-scale low-resolution datasets like WebVid-10M (10.7M 720p videos), is deliberately designed to prioritize quality over quantity, which is critical for high-resolution video generation and finetuning. The specific reasons are as follows:
> - ***High costs of collection and storage.***  We fully acknowledge the concern about the dataset scale (42K short and 17K long pairs). As noted, collecting high-quality UHD (4K/8K) videos with comprehensive structured captions is extremely resource-intensive: each original video undergoes manual search and review to ensure visual quality from the source of video collection (Line-93/95 in Sec. 2.1), and each split video undergoes multi-stage filtering (statistical checks, model-based purification) to ensure semantic alignment. Moreover, the current short version with the original 4K/8K resolution already occupies 1.78TB of storage, which limits the scale in the initial release.
> - ***Bridging the quality gap of open-source data/models.*** Existing large-scale datasets (*e.g.*, Panda-70M [8] and Koala-36M [36]) contain millions of low-resolution (<720p) videos with simplistic captions, which fail to support high-quality model training and native UHD generation. These datasets have the advantage of scale and can be used for model pre-training. In contrast, our newly proposed UHD UltraVideo is committed to filling the gap in high-quality video datasets rather than pursuing sheer quantity for fine-tuning (see Line-57 / Line-68 / Line-463). This is reflected in its visual fidelity, ultra-high resolution, fine-grained captions, and rich topics, and it is mainly used for post-finetuning to further improve model quality.
> - ***Effectiveness validation.*** Our experiments show that even with 42K samples, UltraWan already outperforms baseline extrapolation methods, validating the dataset’s effectiveness (Table 4/5 and Fig. 5/6). We also provide ablation studies (Sec. 4) demonstrating that the curated 42K samples are sufficient to drive meaningful improvements in resolution scalability and semantic control.
> - Considering that over 1,000 users have applied for UltraVideo, we plan to apply to supervisors and external organizations for more resources to support subsequent work in the near future, aiming to further expand the application scope and value of UltraVideo. Specifically, we will expand the quantity by an order of magnitude to reach 0.4M, matching OpenVidHD-0.4M. We will provide control information such as depth, scribble, and layout for each video to support controllable video editing tasks similar to VACE [1r], offer more abundant structured CLIP scores, employ more advanced multimodal and perception models to enhance the quality of curation, among other enhancements.
>
> ---
>
> **Q2: Multi-Modal Conditional Generation Support**
>
> Thanks for your valuable advice.
> 1) Our current focus on text-to-video (T2V) is a strategic choice aimed at first establishing a solid foundation for UHD T2V generation. This area remains underexplored, largely due to the lack of high-quality UHD text-video pairs. That said, the dataset now also supports inputs such as images and audio, enabling image-to-video and audio-to-video tasks. The UltraWan model, trained on text-to-video data, has already validated the effectiveness of UltraVideo (Table 4/5 and Fig. 5/6). By first mastering text-driven UHD generation, we aim to lay the groundwork for integrating additional modalities moving forward.
> 2) We fully agree that supporting multi-modal conditions (*e.g.*, depth, scribble, and layout) would significantly enhance the dataset’s applicability. We will add such control information—including depth, scribble, and layout—for the openly accessible UltraVideo samples to support controllable video editing tasks similar to VACE [1r]. Furthermore, we will synchronously release annotations for these modalities in the expanded dataset mentioned in A1. We appreciate your suggestion.
>
> ---
>
> **Q3: Longer Video Durations**
>
> Thanks for pointing this out. We appreciate your insightful question. Our dataset also includes 17K long videos (≥10s) processed using the same curation pipeline, which we believe can support research on long-video generation in two ways:
> 1) The structured captions (*e.g.*, "camera movement," "style") explicitly describe long-range motion. Specifically, each video has been globally described using Qwen2.5-VL-72B [7], which helps models learn coherent dynamics over extended durations.
> 2) The high-quality visual fidelity can enhance and potentially support the quality of long-video generation. This paper focuses on designing a curation pipeline to obtain short videos, while the 17K long videos, as a derivative dataset, can support the current thriving research on long videos. This is also one of our future research directions, as noted in Line-307 of Sec. 5.
> 3) Additionally, as mentioned in A1 regarding the expanded dataset, we will synchronously expand the long-video datasets to further support the development of related research.
>
> [1r] Jiang, Zeyinzi, et al. "Vace: All-in-one video creation and editing." arXiv, 2025.

---

> > ### Comment · Reviewer_1vTp · 2025-08-07
> >
> > I appreciate the authors' thorough responses. As most of my concerns have been resolved, I will keep my positive evaluation.

---

> > > ### Author Response · Authors · 2025-08-08
> > > **Thank Reviewer 1vTp for the Comments**
> > >
> > > Dear Reviewer 1vTp:
> > >
> > > We are very pleased to have addressed your concerns and thank you for recognizing our work. We commit to incorporating the content you suggested in the revised version. Thank you again for your effort in the review and the discussion!
> > >
> > > Best regards!
> > >
> > > Authors of UltraVideo

---

> ### Comment · Area_Chair_H2No · 2025-08-02
> **Please read author rebuttal**
>
> The author has provided a rebuttal to respond to your comments. Please have a read on the author response and discuss with author if necessary.
>
> Thanks,
>
> Your AC

---

### Official Review · Reviewer_39t2 · 2025-07-03

**Rating:** 5
**Confidence:** 4

**Summary:**

This paper introduces UltraVideo, an open-source dataset for text-to-video generation focusing on 4K/8K resolution with comprehensive, structured captions. It creates a high-quality dataset of 42,000 short videos and 17,000 long videos, curated through a meticulous four-stage pipeline designed to ensure high visual fidelity and eliminate low-quality content. Each video is annotated with ten distinct types of structured captions, averaging over 800 words. The authors also present UltraWAN, a fine-tuned version of the WAN-T2V model, which shows significant improvements in generating 1K and 4K resolution videos.

**Dataset Code Accessibility:**

Yes

**Ethical Considerations:**

No, there are no or only very minor ethics concerns

**Final Justification:**

The rebuttal has addressed most of my concerns. So I would like to keep my original rating for accepting this paper.

**Limitations Weaknesses:**

1. The quality of the data curation pipeline (e.g., the accuracy of the scene splitting) and the accuracy of the generated captions have not been validated by human annotators. This would have provided a stronger guarantee of the dataset's quality.
2. The average caption length of over 800 words, while comprehensive, may pose practical challenges. Many current text-to-video models have text encoders with context length limitations that are shorter than this. While the authors propose a random sampling strategy, a more detailed analysis of how to best utilize or condense these long captions for standard models would make the dataset more readily usable.

**Strengths Contributions:**

1. This paper focuses on the creation of Ultra-High Definition video generation dataset, which is quite important to the academic field.
2. The paper details a well-designed four-stage data curation and captioning pipeline.
3. The authors validate their dataset by fine-tuning the WAN-T2V model, creating UltraWAN-1K and UltraWAN-4K.
4. The paper is well-written, organized, and easy to understand.

---

> ### Author Rebuttal · Authors · 2025-07-30
>
> **We sincerely appreciate the reviewer's positive evaluation of our work and valuable suggestions, which are crucial for enhancing the rigor and practicality of our study. Below, we address each limitation raised and outline specific improvements.**
>
> ---
>
> **Q1: Validating Data Curation Pipeline and Caption Accuracy with Human Annotation**
>
> Thank you for your insightful review.
> - We fully agree that professional human annotators are crucial for improving dataset quality, and we have leveraged all accessible laboratory resources during this work to ensure the quality of the final data. Specifically: each original video undergoes manual search and review from the source of video collection to ensure visual quality (Line-93/95 in Sec. 2.1). This not only guarantees video quality but also reduces the difficulty of subsequent filtering, minimizing the proportion of defective videos. Additionally, we conducted an "analysis of non-compliance" experiment in Sec. 2.5, which concluded that "UltraVideo had a failure rate of 2.3%, significantly lower than the 41.5% failure rate of the popular Koala-36M [36]."
> - From the perspective of experimental results, UltraWan, fine-tuned on UltraVideo, achieves higher quantitative metrics (Table 4) and superior visual effects (Fig.~5), which validates the effectiveness of our data curation. Moreover, it can generate semantically consistent videos, as evidenced by the higher semantic consistency in generated videos shown in Fig. 5 and the overall better quantitative indicators in Table 4—both confirming the accuracy of the captions.
> - Furthermore, we have supplemented an ablation experiment of "Ablation Study on Filtered vs. Unfiltered Data" (refer to Q2 from Reviewer WgAh). Specifically, due to limited computational resources during the rebuttal period, we randomly selected 10K subsets from UltraVideo, unfiltered UltraVideo, OpenVidHD-0.4M [20], and OpenSoraPlan [16] respectively for LoRA fine-tuning at 1K resolution for one epoch, resulting in **UltraWan-1K-10K**, **UltraWan-1K-unfiltered-10K**, **UltraWan-1K-OpenVidHD-10K**, and **UltraWan-1K-OpenSoraPlan-10K**. The comparison results of the VBench subset (as shown in Line-261) are presented in the table below.
>
> | **Model** | **Subject Consistency** | **Background Consistency** | **Aesthetic Quality** | **Imaging Quality** |
> | :---: | :---: | :---: | :---: | :---: |
> | **UltraWan-1K-10K** | 97.05% | 98.25% | 62.28% | 67.53% |
> | **UltraWan-1K-unfiltered-10K** | 96.86% | 98.19% | 62.12% | 67.17% |
> | **UltraWan-1K-OpenVidHD-10K** | 96.71% | 98.07% | 60.25% | 64.92% |
> | **UltraWan-1K-OpenSoraPlan-10K** | 96.84% | 98.15% | 61.58% | 66.15% |
>
> We draw the following conclusions:
>     - Thanks to strict control over video sources and secondary manual preview (Line-94), the quality of the unfiltered sub-dataset is slightly inferior to that of the final filtered dataset (UltraWan-1K-unfiltered-10K *vs.* UltraWan-1K-10K). However, UltraWan-1K-unfiltered-10K still shows significant advantages over UltraWan-1K-OpenVid-10K and UltraWan-1K-OpenSoraPlan-10K, especially in terms of improved image quality.
>     - Through visual comparative analysis of the results, we found that UltraWan-1K-unfiltered-10K has slightly weaker semantic adherence capability compared to the filtered version in some scenarios—especially for subjects involving motion, which are more prone to artifacts. Nevertheless, it still outperforms the results trained by OpenVidHD-10K and OpenSoraPlan-10K, which validates the effectiveness of our curation pipeline.
>
> - Considering that over 1K user applications for UltraVideo/UltraWan have been accumulated, we plan to apply for more resources from supervisors and external institutions to support follow-up work in the near future, aiming to further expand the application scope and value of UltraVideo. Specifically, we will:
>    - Expand the dataset size by an order of magnitude to 0.4M, matching OpenVidHD-0.4M;
>    - Provide control information such as depth, scribble, and layout for each video to support controlled video editing tasks similar to VACE [3];
>    - Use more advanced multimodal and perception models to improve curation quality.
> In this process, we will, based on available resources, strive to increase the proportion of trained human annotators in the data curation workflow (which is also our goal). Meanwhile, we will implement secondary verification of annotations by human annotators (evaluation using quantitative accuracy metric) to ensure the high quality of the finally released data.
>
> ---
>
> **Q2: About Structured Caption**
>
> Thank you for your valuable advice. We will add the following suggestions on how to better utilize structured captions in the revised version:
> - Recent video datasets (*e.g.*, Koala-36M [36]) and video generation methods (*e.g.*, Wan2.1 [34] and HunyuanVideo [15]) have demonstrated that detailed captions are essential for model training and application. Thus, long captions have become a key factor in the development of video datasets. Recent video generation models also primarily support long captions as input: for instance, Wan2.1/2.2 [34] uses umt5-xxl [1r] as the text encoder, while HunyuanVideo [15] employs MLLMs for multimodal encoding. In other words, mainstream video generation models already support long captions as input without being restricted by an average of 824 words.
> - Additionally, structured captions also include short captions, with an average length of 33 words, and samples exceeding 77 words account for less than 0.5%. This makes them compatible with standard language models such as CLIP (which supports a maximum of 77 tokens as input).
> - Furthermore, to fully leverage the potential of structured captions, on the basis of our proposed Random caption sampling strategy in Sec. 3.2, one can integrate a pretrained LLM (*e.g.*, Qwen2.5-7B) to extract core attributes (*e.g.*, "subject," "action," and "lighting") into summaries with a limited number of words.
>
> [1r] Xue, Linting, et al. "mT5: A massively multilingual pre-trained text-to-text transformer." NAACL, 2021.

---

> > ### Comment · Reviewer_39t2 · 2025-08-05
> > **Official Comment by Reivewer 39t2**
> >
> > The rebuttal has addressed most of my concerns. So I would like to keep my original rating for accepting this paper.

---

> > > ### Author Response · Authors · 2025-08-05
> > > **Thank Reviewer 39t2 for the Comments**
> > >
> > > Dear Reviewer 39t2:
> > >
> > > Thank you for recognizing our work. We commit to incorporating the content you suggested in the revised version. Thank you again for your effort in the review and the discussion!
> > >
> > > Best regards!
> > >
> > > Authors of UltraVideo

---

> ### Comment · Area_Chair_H2No · 2025-08-02
> **Please read author rebuttal**
>
> The author has provided a rebuttal to respond to your comments. Please have a read on the author response and discuss with author if necessary.
>
> Thanks,
>
> Your AC

---

### Official Review · Reviewer_WgAh · 2025-07-03

**Rating:** 4
**Confidence:** 4

**Summary:**

A high-resolution and quality text-video dataset, the dataset covers 7 major theme and and contains dense captions for the filtered frames.  A main concern is the dataset size is a bit small, which may only be used for fine-tuning for short steps.

**Dataset Code Accessibility:**

No

**Dataset Code Comments:**

I cannot access the samples in huggingface as it requests my contact information, which may violate the review policy.

**Ethical Considerations:**

No, there are no or only very minor ethics concerns

**Final Justification:**

The paper proposes a high-quality video dataset for generative model training. Although I still have concerns about the dataset size, I understand that collecting such data can be very costly. So, I maintain my score to accept.

**Limitations Weaknesses:**

1. Scale. Although I understand that collecting such data can be very costly, it only contains 42K pairs, which makes it challenging to train or finetune a model for long time. For example, the paper also only finetunes WAN for 1 epoch with LORA-16.

2. Experiments are conducted to shown the effectiveness of finetuning the model on the proposed filtered data. An ablation needs to be done to verify the necessity of the data: finetuning WAN on the unfiltered 1K video dataset. Otherwise, it is unclear whether the performance gains comes from the filtered dataset or the 1K video clip.

3. Will this dataset helps refining long-video generation?

4. Does the dataset contains CLIP score between the texts and the videos? Some methods use CLIP score to select captions instead of randomly choosing one. So, it would be better if such scores are included.

**Strengths Contributions:**

1. It proposes a high-resolution text-to-video dataset, which covers 7 major themes with 108 topics. To filter the data, it applies many detection tools such as PaddleOCR to detect text and RAFT algotithm to filter out videos with too slow or too fast motions.

2. Each video is accompanied with a set of comprehensive captions generated by Qwen2.5-VL-72B.  The 9 catergories of captions can be useful in  training the text-to-video models.

3. It verifies the quality by finetuning WAN on the proposed dataset. Compared to the extrapolated WAN1.3B-1K, the model finetuned on the high-quality dataset is better in term of object and consistency.

---

> ### Author Rebuttal · Authors · 2025-07-30
>
> **Thank you very much for your valuable comments and insights, which have provided important guidance for improving our work. We have carefully considered each point and would like to respond in detail as follows.**
>
> ---
>
> **Q1: Scale of UltraWan**
>
> Thank you for your insightful review, and the main reasons are as follows:
> - ***High costs of collection and storage.***  We fully acknowledge the concern about the dataset scale (42K short and 17K long pairs). As noted, collecting high-quality UHD (4K/8K) videos with comprehensive structured captions is extremely resource-intensive: ***each original video undergoes manual search and review to ensure visual quality*** from the source of video collection (Line-93/95 in Sec. 2.1), and each split video undergoes multi-stage filtering (statistical checks, model-based purification) to ensure semantic alignment. Moreover, the current short version with the original 4K/8K resolution already occupies 1.78TB of storage, which limits the scale in the initial release.
> - ***Bridging the quality gap of open-source data/models.*** Existing large-scale datasets (*e.g.*, Panda-70M [8] and Koala-36M [36]) contain millions of low-resolution (<720p) videos with simplistic captions, which fail to support high-quality model training and native UHD generation. These datasets have the advantage of scale and can be used for model pre-training. In contrast, our newly proposed UHD UltraVideo is committed to filling the gap in high-quality video datasets rather than pursuing sheer quantity for fine-tuning (see ***Line-57/68/463***). This is reflected in its visual fidelity, ultra-high resolution, fine-grained captions, and rich topics, and it is mainly used for post-finetuning to further improve model quality.
> - ***One-epoch fine-tuning is sufficient.*** Just as EMU [1r] improved visual quality by fine-tuning SDXL [2r] with 2K high-quality data, our 1-epoch LoRA training is a pragmatic choice given computational constraints. However, we show that it already outperforms baseline extrapolation methods, validating the dataset’s effectiveness (Table 4/5 and Fig. 5/6).
> - Considering that over 1K user applications for UltraVideo/UltraWan have been accumulated, we plan to apply for more resources from supervisors and external institutions to support follow-up work in the near future, aiming to further expand the application scope and value of UltraVideo. Specifically, we will: ***1)*** expand the quantity by an order of magnitude to 0.4M to match OpenVidHD-0.4M; ***2)*** provide control information such as depth, scribble, and layout for each video to support controlled video editing work similar to VACE [3]; ***3)*** offer richer structured CLIP scores; ***4)*** and use more advanced multimodal and perception models to improve the quality of curation, among other initiatives.
>
> [1r] Dai, Xiaoliang, et al. "Emu: Enhancing image generation models using photogenic needles in a haystack." arXiv, 2023.
>
> [2r] Podell, Dustin, et al. "Sdxl: Improving latent diffusion models for high-resolution image synthesis." ICLR, 2024.
>
> ---
>
> **Q2: Ablation Study on Filtered vs. Unfiltered Data**
>
> Thank you for your valuable advice. The unfiltered 1K video dataset (that is, one that does not include Statistical Data Filtering in Sec. 2.2 and Model-Based Data Purification in Sec. 2.3) contains 62K short videos (Line-120). Due to computational constraints during the rebuttal period, we randomly selected a 10K subset from it for LoRA fine-tuning at 1K resolution for one epoch. Additionally, we also randomly selected 10K subsets from OpenVidHD-0.4M [20] and OpenSoraPlan [16] for this purpose, labeled as **UltraWan-1K-unfiltered-10K**, **UltraWan-1K-OpenVidHD-10K**, and **UltraWan-1K-OpenSoraPlan-10K**, respectively. Results of the VBench subset (Line-261) are compared in below Table.
>
> | **Model** | **Subject Consistency** | **Background Consistency** | **Aesthetic Quality** | **Imaging Quality** |
> | :---: | :---: | :---: | :---: | :---: |
> | **UltraWan-1K-42K** | 97.27% | 98.26% | 62.50% | 67.74% |
> | **UltraWan-1K-10K** | 97.05% | 98.25% | 62.28% | 67.53% |
> | **UltraWan-1K-unfiltered-10K** | 96.86% | 98.19% | 62.12% | 67.17% |
> | **UltraWan-1K-OpenVidHD-10K** | 96.71% | 98.07% | 60.25% | 64.92% |
> | **UltraWan-1K-OpenSoraPlan-10K** | 96.84% | 98.15% | 61.58% | 66.15% |
>
> We draw the following conclusions:
> - Thanks to strict control over video sources and secondary manual preview (Line-94), the quality of the unfiltered sub-dataset is slightly inferior to that of the final filtered dataset (UltraWan-1K-unfiltered-10K *vs.* UltraWan-1K-10K). However, UltraWan-1K-unfiltered-10K still shows significant advantages over UltraWan-1K-OpenVid-10K and UltraWan-1K-OpenSoraPlan-10K, especially in terms of improved image quality. In addition, the metrics of UltraWan-1K-10K also slightly decrease compared to the full-scale model UltraWan-1K-42K.
> - Through visual comparative analysis of the results, we found that UltraWan-1K-unfiltered-10K has slightly weaker semantic adherence capability compared to the filtered version in some scenarios—especially for subjects involving motion, which are more prone to artifacts. Nevertheless, it still outperforms the results trained by OpenVidHD-10K and OpenSoraPlan-10K, which validates the effectiveness of our curation pipeline.
>
> ---
>
> **Q3: Utility for Long-Video Generation**
>
> Thank you for your insightful question. Our dataset also includes 17K long videos (≥10s) processed with the same curation pipeline, which we believe can support long-video generation research in two ways:
> - The structured captions (*e.g.*, "camera movement" and "style") explicitly describe long-range motion, as each video is globally described via Qwen2.5-VL-72B [7]. This helps models learn coherent dynamics over extended durations.
> - The high-quality visual fidelity can enhance and potentially support the quality of long-video generation. This paper focuses on designing a curation pipeline to obtain short videos, while the 17K long videos, as a derivative dataset, can support the current booming research on long videos. This is also one of our future research directions, as noted at Line-307 in Sec. 5.
>
> ---
>
> **Q4: Inclusion of CLIP Scores**
>
> Thank you for the insightful question. Our open dataset provides similarity scores between the Summarized Description and VideoCLIP-XL-v2 [35] (Line-158 in Sec. 2.3). For the other 9 structured captions, the same model can be easily used to compute their respective scores. We accept this suggestion and promise to add the remaining 9 columns of CLIP scores to the relevant open-source code repository in the upcoming updated version.
>
> ---
>
> **Q5: Dataset Accessibility**
>
> Thanks for pointing this out. We apologize for the issue regarding access to samples on Hugging Face, which previously required contact information. The main reason we initially set up access requests with automatic approval was to prevent access by crawler accounts and, at the same time, to obtain a general count of users utilizing the dataset, so as to assess the impact of UltraVideo on the community. We assure you that we do not engage in any selection based on personal information (with over 1,000 user applications for UltraVideo/UltraWan accumulated so far, it is highly unlikely to identify reviewer information). Considering the authors' concerns, we have already disabled this access request requirement during the rebuttal period, allowing anonymous access to dataset samples and the codebase. This complies with review policies and ensures that reviewers can freely view the samples and annotations.

---

> > ### Comment · Reviewer_WgAh · 2025-08-02
> >
> > I thank the authors for their responses. I understand that it is costly to collect more video data, especially for 4k videos. I encourage authors put the ablation results to help researchers understand the use of each procedure in the whole data curation process. I maintain my score to accept.

---

> > > ### Author Response · Authors · 2025-08-02
> > > **Thank Reviewer WgAh for the Comments**
> > >
> > > Dear Reviewer WgAh:
> > >
> > > Thank you for recognizing our work. We commit to incorporating the content you suggested in the revised version. Thank you again for your effort in the review and the discussion!
> > >
> > > Best regards!
> > >
> > > Authors of UltraVideo

---

### Decision · Program_Chairs · 2025-09-18

**Decision:**

Accept (poster)

**Comment:**

This paper receives two accepts and two weak accepts. All reviewers think this paper has value to the video-text generation community by introducing a high-quality and open-sourced video-text dataset. This paper presents a detailed data collection and annotation pipeline and verify the effectiveness of proposed dataset. The AC agrees with reviewers and makes an accept recommendation.

===== FINAL UPDATE FROM DB Track PCs ====

The final decision for this paper has been taken by the program chairs after consultation with the SACs. All Senior Area Chairs have ranked papers according to the feedback from the AC during the review process. We decided to leave the original meta-review to reflect the opinion of the AC in light of the initial discussions with reviewers and SAC.